# Influence of Obesity on Bone Turnover Markers and Fracture Risk in Postmenopausal Women

**DOI:** 10.3390/nu14081617

**Published:** 2022-04-13

**Authors:** Juan J. López-Gómez, José L. Pérez-Castrillón, Isabel García de Santos, María Pérez-Alonso, Olatz Izaola-Jauregui, David Primo-Martín, Daniel A. De Luis-Román

**Affiliations:** 1Department of Endocrinology and Nutrición, Hospital Clínico Universitario de Valladolid, 47003 Valladolid, Spain; olatzizaola@yahoo.es (O.I.-J.); dprimoma@saludcastillayleon.es (D.P.-M.); dadluis@yahoo.es (D.A.D.L.-R.); 2Centro de Investigación Endocrinología y Nutrición (IENVA), University of Valladolid, 47002 Valladolid, Spain; uvacastrv@gmail.com (J.L.P.-C.); joseluis.perez@uva.es (M.P.-A.); 3Department of Internal Medicine, Hospital Universitario Rio Hortega, 47012 Valladolid, Spain; 4School of Medicine, University of Valladolid, 47002 Valladolid, Spain; isa.garcia.desantos@gmail.com

**Keywords:** bone metabolism, obesity, osteoporosis, fracture, bone turnover markers

## Abstract

Background and aims: The relationship between obesity and bone metabolism is controversial. In recent decades, the protective role of obesity in the development of osteoporosis is questioned. The aims of this study are the following: to evaluate the differences in bone turnover markers between postmenopausal women with and without obesity and to compare the risk of fracture at five years between these groups. Methods: An observational longitudinal prospective cohort study of postmenopausal women with obesity (O) (body mass index (BMI) > 30 kg/m^2^) and non-obesity (NoO) (BMI < 30 kg/m^2^) is designed. 250 postmenopausal women are included in the study (NoO: 124 (49.6%) and O: 126 (50.4%)). It measures epidemiological variables, dietary variables (calcium intake, vitamin D intake, smoking, alcohol consumption, and physical activity), biochemicals (β-crosslap, type I procollagen amino-terminal peptide (P1NP), 25OH-vitamin D, and parathyroid hormone (PTH)), anthropometric variables, and fracture data five years after the start of the study. The mean age is 56.17 (3.91) years. Women with obesity showed lower levels of vitamin D (O: 17.27 (7.85) ng/mL, NoO: 24.51 (9.60) ng/mL; *p* < 0.01), and higher levels of PTH (O: 53.24 (38.44–65.96) pg/mL, NoO: 35.24 (25.36–42.40) pg/mL; *p* < 0.01). Regarding the bone formation marker (P1NP), it was found to be high in women without obesity, O: 45.46 (34.39–55.16) ng/mL, NoO: 56.74 (45.34–70.74) ng/mL; *p* < 0.01; the bone resorption marker (β-crosslap) was found to be high in women with obesity, being significant in those older than 59 years (O: 0.39 (0.14) ng/mL, NoO 0.24 (0.09) ng/mL; *p* < 0.05). No differences are observed in the risk of fracture at 5 years based on BMI (OR = 0.90 (95%CI 0.30–2.72); *p* = 0.85). Conclusions: Postmenopausal women with obesity showed lower levels of bone formation markers; older women with obesity showed higher markers of bone resorption.

## 1. Introduction

Bone is an active organ on which many factors act. Osteoporosis and the risk of osteoporotic fracture are pathologies that affect the bone. In recent years, these entities are acquiring great relevance due to the progressive aging of the population and the impact on quality of life and the economy that they have on society [1,2].

Many epidemiological studies have shown that low weight and body mass index (BMI) are indicators of high risk of fracture, just as high weight and BMI are protective factors. Although, recently, this relationship does not seem to be so clear, and it is observed that obesity can be related to certain types of fractures [3,4].

Obesity is the most prevalent metabolic disease in the developed world and is one of the main causes of morbidity and mortality. The prevalence of obesity has tripled between 1975 and 2016 according to WHO data. In 2016, more than 1.9 billion adults were overweight, and more than 650 million people were obese [5].

During the past decades, obesity and osteoporosis have become major health problems, and the belief that obesity protects against osteoporosis has been questioned. In fact, some clinical and epidemiological studies have shown that excess fat mass could be a risk factor for osteoporosis and fragility fractures [6].

The factors related to obesity that negatively influence bone mass are mainly associated with an increase in the percentage of fat mass since obesity is a proinflammatory state that is associated with the secretion of a series of cytokines (IL-6, TNF-α) and adipokines (adiponectin, leptin…). Although the cytokines have been observed to have a negative influence on bone, the role of the adipokines is still partially unknown in humans [7]. On the other hand, in patients with obesity, there is a decrease in the levels of circulating 25OHvitamin D, mostly due to its sequestration by adipose tissue. This situation can produce an alteration in the formation of bone, altering its quantity as well as its quality (architecture). In relation to this last point, the decrease in 25OHvitamin D can be associated in some cases, with an increase in PTH that can independently influence 25OHvitamin D in bone metabolism [8,9].

Increased lean mass or fat-free mass is associated with increased bone mass due to an increased mechanical load on the bone relative to weight and muscle hypertrophy. The positive effect of increased lean mass is attributed to lifestyle factors such as exercise and diet, estrogenic sufficiency, genetic influences, or a combination of these factors. On the other hand, increased muscle mass has an independent effect on fracture risk by reducing frailty and falls related to osteoporotic fracture [10]. In obesity, the concept of sarcopenic obesity should be considered, which entails a relative decrease in muscle mass in the situation of obesity in some individuals. This sarcopenia would be associated with a worse influence of muscle mass on bone, in addition to an increased risk of fracture due to frailty (increased falls). The proinflammatory situation related to this entity could also have a negative influence on the bone [11].

According to the described situation, the effect of obesity on bone health and fracture risk has yet to be determined. For this reason, this study has been proposed to evaluate the influence of obesity on bone metabolism. Given that bone metabolism, osteoporosis, and the risk of fracture are related to many risk factors, it is necessary to use highly selected populations to control possible confounding factors. For this reason, it was decided to evaluate the differences in markers of bone metabolism and the risk of fracture in postmenopausal female patients with and without obesity.

## 2. Materials and Methods

### 2.1. Study Design

An observational prospective longitudinal cohort study has been designed. All patients included in the study were informed and gave their consent.

This study has been carried out according to the ethical principles of the Declaration of Helsinki and has been approved by the Clinical Research Ethics Committee (CEIC) of the East Area of Valladolid with code PI 19-1517. 

### 2.2. Population and Study Period

The study has been carried out from the following two cohorts of postmenopausal women: one cohort was women with obesity defined as BMI > 30 kg/m^2^; the other cohort was women without obesity defined as BMI < 30 kg/m^2^. The patients belonged to the East and West health areas of Valladolid, Spain.

A total of 250 patients were included in the study, including 124 postmenopausal women without obesity and 126 postmenopausal women with obesity, from whom data on anthropometry, food intake, and biochemical parameters related to bone metabolism in the initial assessment were taken; we also took fracture data five years after the start of study (January 2014).

The selection was made based on the following inclusion criteria: being a postmenopausal woman and being under 65 years of age. Exclusion criteria were being older than 65 years; criteria of premature or early menopause; have severe chronic kidney or liver disease and have the following toxic habits: active alcoholism and/or drug abuse.

### 2.3. Variables

#### 2.3.1. Epidemiological Variables

Age, physical activity, and toxic habits such as alcohol consumption and smoking were recorded as follows: Age: it was calculated based on the date of birth and the of entry into the study;Physical activity: habitual physical activity is defined as that with a minimum duration of 30 min of exercise per day or 60 min on two days;Alcohol consumption: It is considered with the intake of more than 5 g per day. We have considered alcohol abuse with an intake of more than 20 g per day;Smoking: This variable was considered with a smoking habit of more than 6 months;Calcium intake in the diet: Calcium intake was considered based on the dairy rations consumed per day. In total, 200 mg of calcium per dairy ration consumed per day were considered. It was considered a dairy ration (1 glass of milk of 200 mL, 2 yogurts, or 1 portion of 100 g cheese);Consumption of vitamin D in the diet: The consumption of vitamin D was considered based on the rations of dairy products consumed per day. In total, 0.2 μg of vitamin D per dairy ration was considered. It was considered a dairy ration (1 glass of milk of 200 mL, 2 yogurts, or 1 portion of 100 g cheese).

#### 2.3.2. Biochemical Variables

Vitamin D, plasma calcium, and bone turnover markers were recorded as follows:Vitamin D: This was determined by electrochemiluminescence immunoassay. Cobas 6000 e-601 (Roche Diagnostics, Basel, Switzerland) with a measurement range between 3.00–70.0 ng/mL;Plasma calcium: Total calcium was determined by the ocresolphthalein Schearzenbach method;Bone turnover markers: Three parameters were collected as bone turnover markers. These were beta-crosslap, type I procollagen amino-terminal propeptide (P1NP) and bone non-specific alkaline phosphatase (FA);
○Beta-Crosslap: This is a marker of bone resorption. The measurement range was 0.010–6.00 ng/mL with a functional sensitivity of 0.07 ng/mL. Normal values are different depending on the stage of life. In the case of a postmenopausal woman, the reference value is 0.556–1.008 ng/mL;○P1NP: This bone formation marker is called the amino-terminal propeptide of type I procollagen. The measurement range was 5–1200 μg/L. The reference value in a postmenopausal woman is <76.3 ng/mL;○Alkaline phosphatase: Total alkaline phosphatase (not bone-specific) was considered. This is a marker of bone formation.


#### 2.3.3. Anthropometric Variables

The anthropometric assessment of the subjects was performed by determining weight, height, and body mass index (BMI).

Weight was measured without clothing with an accuracy of ±0.5 kg using a manual scale to the nearest 0.1 kg (SECA, Birmingham, UK). Height was measured with the patient in an upright position to the nearest centimeter using a stadiometer (SECA, Birmingham, UK). The Body Mass Index (BMI) was calculated using the following formula:BMI = weight (kg)/height (m) × height (m)(1)

In this case, the obesity cut-off with a BMI > 30 kg/m^2^ was used for comparison between groups.

#### 2.3.4. Fracture Variables

Osteoporotic-type fracture: For this type of fracture, classic osteoporotic locations were considered (vertebral compression, femoral neck fracture, and distal radius fracture (Colles fracture));Non-osteoporotic fracture: Those that were not found in the typical osteoporotic locations already described (fractures due to trauma, or low impact fractures in non-classical locations of osteoporosis);Incidental fracture: To detect this type of fracture, the available imaging tests were reviewed in search of fractures that had gone unnoticed.

### 2.4. Statistical Analysis

Data were processed using the SPSS statistical package (SPSS for windows version 15.0, 2008 SPSS INC, Chicago, IL, USA).

Quantitative variables with normal distribution were described as mean and standard deviation (Mean (SD)), quantitative variables with non-normal distribution were described as Median and interquartile range (Median (p25–p75)) and, finally, qualitative variables as total number and percentages (Total number (%)).

The inferential analysis tests used were Student’s *t*-test to compare means of normal quantitative variables; the Mann-Whitney U test to compare means of non-normal variables. Chi-square test to compare qualitative variables. Linear regression analysis to compare continuous variables. Multivariate logistic regression analysis to assess the causal relationships between qualitative variables. The significance level was conventionally set at *p* < 0.05.

## 3. Results

### 3.1. Description of the Sample

A total of 250 postmenopausal women with a mean age of 56.17 (3.91) years and a median body mass index (BMI) of 32.27 (24.14–39.59) kg/m^2^ were analyzed.

The presence or absence of obesity was established as a study parameter, so the comparison will be established based on BMI (patients with a BMI > 30 kg/m^2^ (O) vs. patients with a BMI < 30 kg/m^2^ (NoO)). When we made this division, we had 124 patients (49.6%) with a BMI < 30 kg/m^2^; 126 patients (50.4%) with a BMI of > 30 kg/m^2^.

The differences between the different variables related to bone metabolism between the two cohorts are shown in Table 1. An increase in tobacco consumption was observed in patients with a BMI of less than 30 kg/m^2^ and a decrease in the amount of physical activity among patients with a BMI greater or equal to 30 kg/m^2^. On the other hand, there was an increased dietary intake of calcium and vitamin D in the group of patients with a BMI greater than 30 kg/m^2^ (Table 1).

### 3.2. Differences in Bone Turnover Biochemical Parameters

In obese patients (BMI > 30 kg/m^2^) there is a lower 25OHvitamin D levels (O:17.27 (7.85) ng/mL; NoO: 24.51 (9.60); *p* < 0.01), and a higher in intact PTH levels (O:53.24 (38.44–65.96) ng/mL; NoO: 35.24 (25.36–42.40) ng/mL *p* < 0.01). It was also observed to have lower P1NP levels (O: 45.46 (34.39) ng/mL; NoO: 56.74 (45.34–70.74) ng/mL; *p* < 0.01), a marker of bone formation, and we did not observe an effect on beta-crosslaps (O:0.34 (0.14) ng/mL; NoO: 0.33 (0.14) ng/mL; *p* > 0.05).

A stratified analysis by age quartiles was performed (Q1: under 53 years: 89 (35.6%); Q2: 53–56 years: 69 (27.6%); Q3: 56–59 years: 65 (26%); Q4: older than 60 years: 27 (10.8%) patients). The differences in the parameters according to the quartiles are shown in Table 2. A lower vitamin D value and a higher PTH maintained in the four quartiles were observed in patients with BMI > 30 kg/m^2^. In these patients, P1NP was observed to be lower in the lowest age quartiles, while crosslaps were found to be high in the highest age quartile Table 2.

### 3.3. Correlation between BMI and Bone Turnover Biochemical Markers

A negative correlation was observed between the vitamin D levels and P1NP, while a positive correlation was observed between PTH and body mass index (Table 3).

When stratifying according to age quartiles, a negative correlation was observed that was maintained in all quartiles of vitamin D and in the youngest age quartiles of P1NP. PTH maintained the correlation in all quartiles except in the highest age quartile, and the resorption parameter showed a correlation in the third age quartile (Table 3).

### 3.4. Multivariate Analysis

No significant differences were observed in fractures at five years of any type according to BMI (Figure 1).

A multivariate analysis was performed based on age, body mass index, and risk factors for fracture (age, smoking, physical activity, and alcohol consumption) to assess the risk factors for fracture without detecting data on the relationship between BMI and risk of total fracture risk (OR = 0.90 (95%CI 0.30–2.72); *p* = 0.85) (Osteoporotic Fracture: OR = 1.61 (95%CI 0.17–15.06); *p* = 0.68. Non-osteoporotic Fracture: OR = 0.77 (95%CI 0.23–2.65), *p* = 0.68).

## 4. Discussion

In the study carried out, lower levels of 25OHvitamin D were observed in obese women regardless of age. When evaluating bone metabolism parameters, a higher bone formation marker was observed in younger non-obese postmenopausal women and a higher bone resorption marker (β-crosslap) in older obese postmenopausal women.

An inverse relationship was found between BMI and vitamin D levels, such that obese women had less circulating vitamin D, especially in 59-year-old women, together with high intact PTH values compared to women who were not obese. This effect may be related to vitamin D, which has an inhibiting influence on PTH [8].

The most likely mechanism may be the dilution of vitamin D (as it is a fat-soluble vitamin) in the large fat deposits, decreasing its concentration in the blood [8]. The basis of this dilution is found in the sequestration of vitamin D by adipose tissue [12]. Other mechanisms may be low sun exposure, poor nutrition, or a decrease in the 25-hydroxylation of vitamin D in the liver, but they can also play a role, along with other factors such as inflammation or insulin resistance. However, there is no evidence that these low levels of vitamin D that occur in people with obesity have consequences for bone tissue, although they may have effects on other organs [9].

In obese postmenopausal women, higher resorption markers and lower bone formation markers were observed compared to non-obese postmenopausal women. Regarding the bone resorption marker (β-crosslap) in non-obese women, lower values were found compared to obese women, although it was only significant in older postmenopausal women. The role of age is very important in this sample since they are patients under 65 years of age and this stratification allows us to properly categorize risk. In fact, according to the NHANES study in its 2011 to 2018 cutoff, from the age of 60, the increase in fat mass and the decrease in lean mass are associated with a loss of bone mineral density [13].

The cause could be that in women who were not obese and who were in an early stage of menopause, the estrogen level had not dropped enough to reduce bone formation, since as Cui et al. demonstrated in their study, the significant decrease in bone mass occurs from 45–49 to 55–59 years of age [14]. This evidence contradicts obesity protects bone tissue [6], although obese people maintain circulating estrogens due to peripheral aromatization of androgens in relation to increased fat mass [10,15]. However, estrogen levels are not the only regulators of bone mass and, in fact, in studies such as the one carried out by Corina et al., the estrogenic action of adipose tissue was not observed to have a significant effect on bone, especially in the case of postmenopausal patients [16]. In addition, the adipose tissue also secretes proinflammatory cytokines that could interfere with the balance between bone resorption and formation [3].

No differences were found in the risk of fracture at five years (with a total of 10.30% in the obese versus 10.50% in the non-obese). Regarding the osteoporotic type, it was 2.6% in obese women versus 1.60% in non-obese women. The reason for not finding differences in the risk of fracture between obese and non-obese women could be that there are already alterations at the metabolic level in the bone tissue but that there is not yet a sufficient degree of disease to increase the risk of fracture in obese women [17].

Along these lines, a meta-analysis of 25 cohorts of women aged 63 years and older showed that osteoporotic fractures are less frequent in obese women, but, however, wrist fractures are more frequent compared to non-obese women. No association was found with respect to the most distal part of the lower extremity, but it is thought that these differences in location are related to the pattern of falls, the mechanical force induced by the fall, and that the low BMI of the controls could have masked the fracture risk associated with obesity [18]. However, Adachi et al. showed that obesity does not protect against osteoporotic fractures and even increases the risk of ankle and femur fractures [19].

On the other hand, it has been seen that the distribution of fat mass may be important for bone health, since in two meta-analyses it was observed that abdominal obesity, related to visceral adiposity, was associated with a higher rate of hip fractures [20,21].

These findings can have some implications for the management of these patients. In the first place, it is important to know all the parameters that may affect bone health to plan a therapeutic approach and prevent this bone loss. It is necessary to know how the supplementation of vitamin D and the consumption of calcium and phosphorus may affect this [5]. On the other hand, this bone turnover environment could be affected by weight loss strategies in patients with obesity, and it could increase bone resorption parameters, as it was shown in a study by our group where significant weight loss was associated with an increase in beta-crosslaps [22].

Regarding the main limitations of the study, having chosen two cohorts of women under 65 years of age with recent menopause interferes with the adequate assessment of the risk of fracture, since the negative influence of the decrease in estrogen has not been sufficient. Another associated limitation could be having chosen a short follow-up time (5 years) since a greater number of fractures could have been found in a longer-term follow-up. Another limitation is the lack of measures of adiposity in these patients with body composition, waist and hip, or diabetes presence or absence in all patients. These points could influence over the bone metabolism, and they can be parameters to measure in other studies. Lastly, not having imaging studies that indicate bone mineral density limits the adequate evaluation of the influence on bone metabolism.

In view of the results, there are several possible lines of research, such as the effect of the circulating estrogens in obese postmenopausal women on bone tissue and their functionality, since there are not many studies that identify this functionality.

Likewise, it would also be interesting to carry out a long-term prospective study on the net detrimental or beneficial effect of obesity on bone mass (using imaging tests) and the adequate categorization of fracture risk. It is necessary for the treatment strategies to prevent this bone loss and the possible fracture risk.

## 5. Conclusions

Obese postmenopausal women showed lower marker levels of bone formation, especially at younger ages. Older, obese women showed higher markers of bone resorption. This situation may be related to the fact that obese patients showed a decrease in vitamin D levels regardless of age, which is associated with a high PTH. However, an increased risk of fracture at five years was not found among obese patients.

The interaction of obesity and bone metabolism is complex due to the multitude of factors that interfere with it. The belief that obesity protects against osteoporosis is not a completely clear concept, and more studies are required to evaluate its relationship, determine its incidence, and be able to propose measures to prevent its negative influence on bone.

## Figures and Tables

**Figure 1 nutrients-14-01617-f001:**
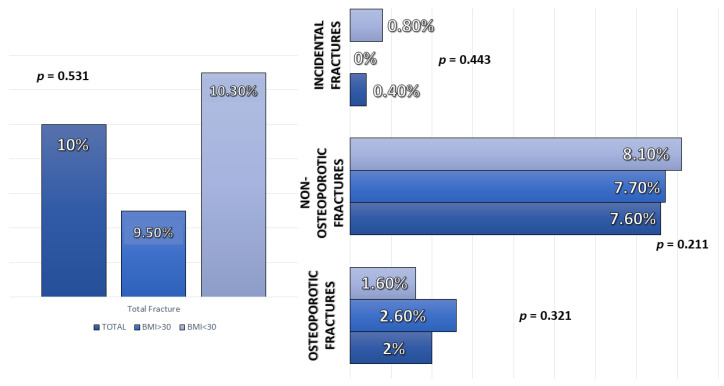
Differences in type fracture depending on body mass index (BMI).

**Table 1 nutrients-14-01617-t001:** Differences between the presence or absence of obesity in variables than may affect in bone metabolism.

	TOTAL	BMI > 30	BMI < 30	*p*-Value
BMI ^1^ (kg/m^2^)	32.27 (24.14–39.59)	39.49 (5.13)	24.14 (2.87)	*p* < 0.01
Age (years)	56.17 (3.91)	56.99 (4.48)	55.33 (3.01)	*p* < 0.01
Alcohol (%)	18 (7.2%)	13 (10.3%)	5 (4%)	0.055
Smoking (%)	59 (23.6%) Exfum 27 (10.8%)	18 (14.4%) 4 (3.2%)	41 (33.1%) 23 (18.5%)	<0.01
Physical activity (%)	144 (57.6%)	38 (30.2%)	106 (85.5%)	<0.01
Dietary calcium (mg)	742.50 (330.18)	857.50 (302.16)	633.52 (322.53)	<0.01
Dietary vitamin D3 (μg)	1.05 (1.39)	1.58 (1.95)	0.63 (0.32)	<0.01

^1^ BMI: Body Mass Index.

**Table 2 nutrients-14-01617-t002:** Differences in bone metabolism biochemical markers between BMI agruped by age Quartiles.

	Q1 (<53 Years)		Q2 (53–56 Years)		Q3 (56–59 Years)		Q4 (>60 Years)	
Variable	BMI ^1^ > 30	BMI ^1^ < 30	BMI ^1^ > 30	BMI ^1^ < 30	BMI ^1^ > 30	BMI ^1^ < 30	BMI ^1^ > 30	BMI ^1^ < 30
Calcium (mg/dL)	9.4 (9.15–9.59)	9.48 (9.27–9.66)	9.5 (9.28–9.77)	9.43 (9.24–9.67)	9.60 (9.50–9.90)	9.45 (9.29–9.67)	9.40 (9.30–9.67)	9.71 (9.62–9.81)
25OHvitamin D (ng/mL)	** *17.30 (7.46)* **	** *23.56 (10.35)* **	** *16.59 (8.86)* **	** *25.66 (8.95)* **	** *18.24 (8.66)* **	** *24.39 (9.54)* **	** *16.48 (6.01)* **	** *26.42 (7.21)* **
PTH (pg/mL)	** *54.36 (40.77–64.44)* **	** *36.23 (28.74–42.01)* **	** *51.89 (37.34–69.52)* **	** *29.9 (24.72–39.90)* **	** *51.27 (38.68–68.96)* **	** *36.39 (24.04–46.04)* **	54 (38.06–60.55)	48.13 (31.18–48.18)
P1NP (ng/mL)	** *46.11 (29.91–55.81)* **	** *65.49 (50.64–85.55)* **	** *43.40 (32.86–46.40)* **	** *56.04 (44.78–70.01)* **	48.35 (37.89–62.08)	47.84 (40.55–56.83)	43.49 (34.91–59.63)	61.16 (30.42–77.01)
CROSSLAPS (ng/mL)	0.32 (0.16)	0.38 (0.17)	0.32 (0.11)	0.30 (0.98)	0.35 (0.13)	0.29 (0.12)	** *0.39 (0.14)* **	** *0.24 (0.09)* **
Alcaline Phosphatase (mg/dL)	** *77.10 (23.66)* **	** *92.65 (20.96)* **	75.92 (17.80)	81.21 (25.79)	83.61 (20.97)	82.39 (17.19)	75.83 (21.21)	85.66 (29.29)

^1^ BMI: Body Mass Index; PTH: Parathyroid Hormone; P1NP: amino-terminal propeptide of type I procollagen. ***p-value < 0.05 (bold and italics)***.

**Table 3 nutrients-14-01617-t003:** Correlation analysis of body mass index (BMI) with bone metabolism parameters in total sample and grouped by age quartiles.

	Calcium (mg/dL)	25OHvitamin D (ng/mL)	PTH ^1^ Intact (pg/mL)	P1NP1 ^2^ (ng/mL)	Crosslaps (ng/mL)	Alcaline Phosphatase (mg/dL)
TOTAL	r = 0.09 *p* = 0.17	** *r = −0.39* ** ** *p < 0.01* **	** *r = 0.52* ** ** *p < 0.01* **	** *r = −0.29* ** ** *p < 0.01* **	r = 0.04 *p* = 0.24	** *r = −0.13* ** ** *p = 0.04* **
Q1 (<53 years)	r = −0.22 *p* = 0.04	** *r = −0.34* ** ** *p < 0.01* **	** *r = 0.55* ** ** *p < 0.01* **	** *r = −0.37* ** ** *p < 0.01* **	r = −0.128 *p* = 0.24	** *r = −0.27* ** ** *p = 0.01* **
Q2 (53–56 years)	r = 0.03 *p* = 0.81	** *r = −0.43* ** ** *p < 0.01* **	** *r = 0.53* ** ** *p < 0.01* **	** *r = −0.39* ** ** *p < 0.01* **	r = 0.04 *p* = 0.77	r = −0.06 *p* = 0.63
Q3 (56–59 years)	r = 0.23 *p* = 0.08	** *r = −0.35* ** ** *p < 0.01* **	** *r = 0.51* ** ** *p < 0.01* **	r = 0.08 *p* = 0.54	** *r = 0.27* ** ** *p = 0.03* **	r = 0.06 *p* = 0.65
Q4 (>59 years)	r = −0.14 *p* = 0.49	** *r = −0.47* ** ** *p = 0.01* **	r = 0.32 *p* = 0.23	r = −0.20 *p* = 0.31	r = 0.11 *p* = 0.59	r = −0.01 *p* = 0.96

^1^ PTH: Parathyroid Hormone; ^2^ P1NP: amino-terminal propeptide of type I procollagen. ***p-value < 0.05 (bold and italics)***.

## Data Availability

Not applicable.

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
