# Peer review of "Influence of Obesity on Bone Turnover Markers and Fracture Risk in Postmenopausal Women"

_nutrients, 2022, doi:10.3390/nu14081617_

Round 1

Reviewer 1 Report

Here the authors report on an observational longitudinal prospective study involving obese and non-obese postmenopausal women, in which they aimed to assess the influence of obesity on bone turnover markers and fracture risk. They found that vitD levels were lower and PTH higher in obese vs non-obese women. P1NP was higher in non-obese women, while CTX were higher in obese elder ones. They did not find a difference in the risk of fracture at 5 years based on the BMI.

We have some comments and questions: as the authors say, the lack of info regarding BMD in this cohort is a limitation of the study; we think this is an important point, indeed. Also, there is no info regarding the diabetic condition in this cohort, which is relevant, too. Five years after patients' enrollment, the authors take fracture data; we think that it would have been important to have also a follow-up on anthropometry, life style and biochemical measures. A missing information is the number of years since menopause, which may relate to several parameters in this study.

Regarding alcohol consumption, which is indicated as exclusion criterium, clarify line 108: is that the max daily consumption?

In the Discussion, lines 238-246 (and similar sentences throughout the results section), we think the terms "higher" or "lower" are more appropriate than "increased" or "decreased" (which sound as referred to a basal condition). 

Please revise all the language throughout the text, since there are several typos (e.g. line 177 consumtion; line 193 an stratified; line 196 in th four quartiles; in several points, kg/m2), words or acronyms that are not in English (e.g. Headings in Table 1,2 and 3), and sentences that need rephrasing (e.g., line 52; line 246; lines 266-268). 

"General cytokine" is not a nice term; please delete "general". Adipokine and inflammatory cytokine measurement would have been of interest and relevant to define the characteristics of the two subgroups of patients.

Author Response

Dear reviewers and editorial office:

First, I would like to thank you for the trust placed in our group by reviewing and considering our article.

According to the comments received, we have made a series of corrections in our article that I list below:

Reviewer 1:

Here the authors report on an observational longitudinal prospective study involving obese and non-obese postmenopausal women, in which they aimed to assess the influence of obesity on bone turnover markers and fracture risk. They found that vitD levels were lower and PTH higher in obese vs non-obese women. P1NP was higher in non-obese women, while CTX were higher in obese elder ones. They did not find a difference in the risk of fracture at 5 years based on the BMI.

We have some comments and questions:

- As the authors say, the lack of info regarding BMD in this cohort is a limitation of the study; we think this is an important point, indeed. Also, there is no info regarding the diabetic condition in this cohort, which is relevant, too. Five years after patients' enrollment, the authors take fracture data; we think that it would have been important to have also a follow-up on anthropometry, lifestyle and biochemical measures. A missing information is the number of years since menopause, which may relate to several parameters in this study.

  • As the reviewers comments it could be interesting to know the changes in lifestyle and anthropometry in five years after the measure. However, we only study the presence of fracture in our registry. On the other hand, the presence or absence of diabetes haven’t been measured. All of these points are being added in limitations of the study.

Line 309-311: “Other limitation is the lack of measure of waist and hip, or diabetes presence or absence in all patients, these points could influence over the bone metabolism and it can be pa-rameters to measure in other studies”.

Respect to the years of menopause we haven’t showed in data but premature menopause was an exclusion criteria of the study and it has been written in the article.

Line 97-100: “Exclusion criteria were being older than 65 years; criteria of premature or early menopause; have severe chronic kidney or liver disease and have toxic habits: active alcoholism and/or drug abuse.”

  • Regarding alcohol consumption, which is indicated as exclusion criterium, clarify line 108: is that the max daily consumption?
  • We consider alcohol abuse as consume of more than 20 g alcohol per day as exclusion criteria. However, to consider alcohol consumption we have considered more than 5 g/day, alcohol consume 5-20 g/alcohol day and alcohol abuse more than 20 g alcohol per day.

Line 108-109: Alcohol consumption: It is considered with the intake of more than 5 g per day. We have considered alcohol abuse with the intake of more than 20 g per day. 

  • In the Discussion, lines 238-246 (and similar sentences throughout the results section), we think the terms "higher" or "lower" are more appropriate than "increased" or "decreased" (which sound as referred to a basal condition).
  • We have changed all this sentences as you recommend.

Line 22-31: “The mean age was 56.17 (3.91) years. Women with obesity showed lower levels of vitamin D (O: 17.27 (7.85) ng/ml, NoO: 24.51 (9.60) ng/ml; p<0.01), and an increase inhigher levels of PTH (O: 53.24 (38.44-65.96) pg/ml, NoO: 35.24 (25.36-42.40) pg/ml; p<0.01). Regarding the bone formation marker (P1NP), it was found to be increased high in women without obesity, O: 45.46 (34.39-55.16) ng/ml, NoO: 56.74 (45.34-70.74) ng/ml; p<0.01; the bone resorption marker (-crosslap) was found to be increased high in women with obesity, being significant in those older than 59 years (O: 0.39 (0.14) ng/ml, No O 0.24 (0.09) ng/ml; p<0.05). No differences were observed in the risk of fracture at 5 years based on BMI (OR=0.90 (95% CI 0.30-2.72); p=0.85). Conclusions: Postmenopausal women with obesity showed decreased lower levels of bone formation markers; and older women with obesity showed increased higher markers of bone resorption.”

Line 188-193: “In obese patients (BMI > 30 kg/m2) there is a lower 25OHvitamin D levels (O:17.27(7.85) ng/ml; NoO: 24.51 (9.60); p<0.01), and a higher in intact PTH levels (O:53.24(38.44-65.96) ng/ml; NoO: 35.24 (25.36-42.40) ng/ml p<0.01). It was observed also a lower P1NP levels (O: 45.46 (34.39) ng/ml; NoO: 56.74 (45.34-70.74) ng/ml; p<0.01), a marker of bone formation, and we did not observe effect on beta-crosslaps(O:0.34 (0.14) ng/ml; NoO: 0.33 (0.14) ng/ml; p>0.05).

A stratified analysis by age quartiles was performed (Q1: Under 53 years: 89 (35.6%); Q2: 53-56 years: 69 (27.6%); Q3: 56-59 years: 65 (26%); Q4: older than 60 years: 27 (10.8%) patients). The differences in the parameters according to the quartiles are shown in table 2. A lower vitamin D values and a higher PTH maintained in the four quartiles were observed in patients with BMI > 30 kg/m2 In these patients, P1NP was observed to be lower in the lowest age quartiles, while crosslaps were found to be high in the highest age quartile table 2.“

Line 239-243: “regardless of age. When evaluating bone metabolism parameters, a significant increase in higher bone formation markers was observed in younger non-obese postmenopausal women and a higher significant in bone resorption markers (-crosslap) in older obese postmenopausal women.

An inverse relationship was found between BMI and vitamin D levels, such that obese women had less circulating vitamin D, especially in 59 years old women, together with a high in intact PTH values compared to women who were not obese. This effect may be related to the fact that vitamin D has an inhibiting influence on PTH [8].”

Line 254-256: “However, there is no evidence that these low levels of vitamin D that occur in people with obesity have consequences on bone tissue, although they may have effects on other organs [9].”

Line 257-264: “In obese postmenopausal women, a higher resorption marker and a lower bone formation marker as observed compared to non-obese postmenopausal women. Regarding the bone resorption marker (-crosslap) in non-obese women, a lower value was found compared to obese women, although it was only significant in older postmenopausal women.”

Line 325-328: Obese postmenopausal women showed lower marker levels of bone formation, especially at younger ages. Older, obese women showed higher markers of bone resorption. This situation may be related to the fact that obese patients showed a decrease in vitamin D levels regardless of age, associated with a high PTH. However, an increased risk of fracture at five years was not found among obese patients.

  • Please revise all the language throughout the text, since there are several typos (e.g. line 177 consumtion; line 193 an stratified; line 196 in th four quartiles; in several points, kg/m2), words or acronyms that are not in English (e.g. Headings in Table 1,2 and 3), and sentences that need rephrasing (e.g., line 52; line 246; lines 266-268).
  • We have corrected the sentences that you mentioned and the mistakes in tables and figures. Sorry for the inconvenience.
  • "General cytokine" is not a nice term; please delete "general". Adipokine and inflammatory cytokine measurement would have been of interest and relevant to define the characteristics of the two subgroups of patients.
  • In first place, we have deleted the term “general” from throughout the text.

Line 51-55: “The factors related to obesity that negatively influence bone mass are mainly as-sociated with an increase the percentage of fat mass, since obesity is a proinflammatory state that is associated with the secretion of a series of cytokines (IL-6, TNF-) and adipokines (adiponectin, leptin…). Although the cytokines have been observed to have negative influence on bone, the role of the adipokines is still partially unknown in humans [7].”

  • On the other hand, it would have been very interesting to study the role of citokynes and adipokynes on bone metabolism, but we haven’t measured this parameters.

Reviewer 2 Report

The work was done competently and the manuscript is very well-written, yet a number of points and concerns exist that should be dealt with effectively by the authors prior to any consideration. These points are provided below:

Minor comments

  1. A few linguistic errors require attention at the grammatical and syntactical level. Appropriate revisions should be made by the authors. Representative points requiring attention: e.g. headers in table 1
  2. The discussion is relatively short and can be modified accordingly in order for the reader to follow the plethora of information. A summary with the key findings can also be added.
  3. In the spirit of the previously explained way of writing and presenting the experimental

results, the visualization of the results (all figures) can be further improved (strong suggestion).

Major comments

  1. In the manuscript, the authors conclude that obese women showed decreased levels of bone formation markers. Although this conclusion is supported by the analysis, and given that several studies support the fact that obese women project lower possibility of presenting with osteoporosis, I suggest that other markers should also be introduced in the study. This comment is based on the fact that several studies support the fact that Visceral Adiposity Index (VAI) has been proven to be a direct marker of adipose distribution. The authors should take into account fat distribution as well (as a marker of increased risk for inflammation that has a severe impact on vitamin D metabolism which in turn is closely associated with the onset of osteoporosis).
  2. To my understanding, general adiposity index, referred to as BMI (body mass index), with abdominal obesity indices, such as waist circumference, waist-to-hip ratio, and waist-to-height ratio, in order to unravel the most efficient predictive marker should also be included in the study in order to have more clear view regarding the aforementioned findings and further delineate the impact of adipose tissue.

Author Response

Dear reviewers and editorial office:

First, I would like to thank you for the trust placed in our group by reviewing and considering our article.

According to the comments received, we have made a series of corrections in our article that I list below:

Reviewer 2:

The work was done competently and the manuscript is very well-written, yet a number of points and concerns exist that should be dealt with effectively by the authors prior to any consideration. These points are provided below:

Minor comments

A few linguistic errors require attention at the grammatical and syntactical level. Appropriate revisions should be made by the authors. Representative points requiring attention: e.g. headers in table 1

  • We have revised this grammatical and syntactical errors. Some of these errors have been a lack of attention. Sorry about that.

The discussion is relatively short and can be modified accordingly in order for the reader to follow the plethora of information. A summary with the key findings can also be added.

  • We have added some paragraphs about importance of these findings and future investigations lines. We have also clinical implications of these findings in order to clear out the aims of the study.

Line 296-303: “These findings can have some implications for the management of these patients. In first place it is important to know all the parameters that may affect bone health to plan a therapeutic approach and prevention of this bone loss. It is necessary to know how the supplementation of vitamin D, and the consumption of calcium and phos-phorus may affect to this [5]. On the other hand, this bone turnover environment could be affected by weight loss strategies in patients with obesity and it could increase bone resorption parameters as it’s shown in study by our group where a significant weight loss was associated to an increase in beta-crosslaps [22].”

In the spirit of the previously explained way of writing and presenting the experimental

results, the visualization of the results (all figures) can be further improved (strong suggestion).

  • The results showed in tables are so many that it is difficult to show as figures. However, we have changed the figure and we have changed the redaction of results for a better understanding.

Major comments

In the manuscript, the authors conclude that obese women showed decreased levels of bone formation markers. Although this conclusion is supported by the analysis and given that several studies support the fact that obese women project lower possibility of presenting with osteoporosis, I suggest that other markers should also be introduced in the study. This comment is based on the fact that several studies support the fact that Visceral Adiposity Index (VAI) has been proven to be a direct marker of adipose distribution. The authors should take into account fat distribution as well (as a marker of increased risk for inflammation that has a severe impact on vitamin D metabolism which in turn is closely associated with the onset of osteoporosis).

  • In this study compare females with and without obesity and study classical factors related to osteoporosis as physical activity, smoking habit, alcohol consumption or calcium and vitamin D consumption. Unfortunately, data from adipose mass and adipose distribution as hip and waist or bioelectrical impedanciometry only has been measured in obese females. It is due to these patients have entered in another study of lifestyle modifications. We cannot include this data because not all the patients have all the measurements.

We have included this point at limitations of the study.

Line 309-311: “Other limitation is the lack of measure of adiposity in this patients with body compo-sition, waist and hip, or diabetes presence or absence in all patients, these points could influence over the bone metabolism, and it can be parameters to measure in other studies.”

To my understanding, general adiposity index, referred to as BMI (body mass index), with abdominal obesity indices, such as waist circumference, waist-to-hip ratio, and waist-to-height ratio, in order to unravel the most efficient predictive marker should also be included in the study in order to have more clear view regarding the aforementioned findings and further delineate the impact of adipose tissue.

  • This study was designed to compare patients with and without obesity, as we mentioned previously unfortunately, we haven’t measured hip and waist circumference. We only measured these parameters in obese women. When we have done the analysis there were no differences at this point and not all the patients have the measure of it, so we think we can’t include this data.

We have added this point at limitations of the study and we think it study of body composition and paper of sarcopenic obesity in this patients could be a future line of investigation.

Line 309-311: “Other limitation is the lack of measure of adiposity in this patients with body composition, waist and hip, or diabetes presence or absence in all patients, these points could influence over the bone metabolism, and it can be parameters to measure in other studies.”

Round 2

Reviewer 1 Report

We thank the authors for having discussed all the points raised by this Reviewer and modified the text accordingly.